# Enablers and barriers to the implementation of socially assistive humanoid robots in health and social care: a systematic review

Irena Papadopoulos,[1] Christina Koulouglioti,[1,2] Runa Lazzarino  ,[1] Sheila Ali[1]

[1]Research Centre for Transcultural Studies in Health, Middlesex University, London, UK
[2]Research and Innovation, Western Sussex Hospitals NHS Foundation Trust, Worthing, UK

**Correspondence to**
Professor Irena Papadopoulos;
r.papadopoulos@mdx.ac.uk

## ABSTRACT

**Objectives** Socially assistive humanoid robots are considered a promising technology to tackle the challenges in health and social care posed by the growth of the ageing population. The purpose of our study was to explore the current evidence on barriers and enablers for the implementation of humanoid robots in health and social care.

**Design** Systematic review of studies entailing hands-on interactions with a humanoid robot.

**Setting** From April 2018 to June 2018, databases were searched using a combination of the same search terms for articles published during the last decade. Data collection was conducted by using the *Rayyan* software, a standardised predefined grid, and a risk of bias and a quality assessment tool.

**Participants** Post-experimental data were collected and analysed for a total of 420 participants. Participants comprised: older adults (n=307) aged ≥60 years, with no or some degree of age-related cognitive impairment, residing either in residential care facilities or at their home; care home staff (n=106); and informal caregivers (n=7).

**Primary outcomes** Identification of enablers and barriers to the implementation of socially assistive humanoid robots in health and social care, and consequent insights and impact. Future developments to inform further research.

**Results** Twelve studies met the eligibility criteria and were included. None of the selected studies had an experimental design; hence overall quality was low, with high risks of biases. Several studies had no comparator, no baseline, small samples, and self-reported measures only. Within this limited evidence base, the enablers found were enjoyment, usability, personalisation and familiarisation. Barriers were related to technical problems, to the robots' limited capabilities and the negative preconceptions towards the use of robots in healthcare. Factors which produced mixed results were the robot's human-like attributes, previous experience with technology and views of formal and informal carers.

**Conclusions** The available evidence related to implementation factors of socially assistive humanoid robots for older adults is limited, mainly focusing on aspects at individual level, and exploring acceptance of this technology. Investigation of elements linked to the environment, organisation, societal and cultural milieu, policy and legal framework is necessary.

## Strengths and limitations of this study

► This review is the first to date focusing on the issues related to the pragmatic implementation of socially assistive humanoid robots in health and social care settings catering to the needs of older adults.
► Quality assessment of the included studies was based on two combined tools to account for the heterogeneity of the underlying study designs.
► Three authors were involved in critical steps of the review (article selection, data extraction, quality assessment of the included studies), and this constitutes a strength of this study.
► The heterogeneity between studies on key issues, such as participants' cognitive health and residential context, study designs and outcomes, prevents quantitative synthesis and hampers consistent assessment of the implementation of socially assistive humanoid robots in health and social care.

**PROSPERO registration number** CRD42018092866.

## INTRODUCTION
### Rationale

The current global landscape in health and social care is highly challenging, demanding innovative and effective actions from policy makers and service providers. For example, it is projected that by 2050 the world's population over the age of 60 years will be about two billion, an increase of 900 million from 2015.[1] Shortages of healthcare professionals and a growing ageing population place enormous pressures onto the health and social care systems of many countries. Older adults are living longer with chronic problems and/or disabilities. At the same time, the size of formal and informal healthcare workforce is shrinking.

The use of artificial intelligence (AI) and robotics provides a major opportunity towards meeting some of the care needs of older adults.[2 3] An advanced form of AI is the one used in socially assistive humanoid robots

(SAHRs). These robots use gestures, speech, facial recognition, movements and, in general, social interaction to assist their users.[4] The robot's goal is to create close and effective interaction with the human user for the purpose of giving assistance and achieving measurable progress in convalescence, rehabilitation, learning and well-being.

In a systematic review of the literature about the use of different available technologies directed to assist older adults, robotic devices and robots were viewed as an encouraging technology that can assist and prolong older adults' independent living.[5] Corroborating this finding, a few additional reviews of the literature have indicated that: (i) SAHRs could have multiple roles in the care of older adults such as in affective therapy and cognitive training[6] and (ii) they could be beneficial in reducing anxiety, agitation, loneliness and improving quality of life, engagement and interaction (especially when used as a therapeutic tool when caring for patients with dementia).[7–9] In addition, reviews related to the acceptance of robots have found it being influenced by numerous factors, such as the perceived need for the technology, the user's previous experiences with it, age, level of education, expectations about what the technology can do, attitudes and cultural background;[10] in fact, robots that were programmed to use verbal and non-verbal communication familiar to the user and to their cultural background were more easily accepted by users.[11] Furthermore, a review of qualitative studies on older adults' experiences with socially assistive robots revealed the complexity of issues associated with their use with older adults, and how these impacted on their attitudes towards robots.[12] For example, issues related to the 'role' that the robot could acquire and to the nature of the human–robot interaction (HRI) revealed a mixture of opinions and emotions. Parallel enquires among health and social care professionals have identified various areas where humanoid and animal-like robots can be helpful, but reported mixed views about their use in healthcare settings, raising issues of staff and patients' safety, and the protection of their privacy.[13] On a similar note, a recent qualitative exploration among different stakeholders in the healthcare context revealed that ethical and legal challenges, the lack of interests from professionals and patients, and concerns related to the robot's appearance and robotic expectations were major barriers to their potential use.[14] Frennert et al's review[15] focused mainly on concerns that need attention when considering the social robots and older adults interface, and urged developers to adopt a more pragmatic and realistic idea of an older adult. Their recommendations addressed the inclusion of older adults in the development process, without considering them incapable of expressing their needs and offering possible solutions to their own problems.

All current reviews shed some light on certain aspects of this complicated relationship: older adults and socially assistive robots. However, in order to effectively meet the care needs of an ageing population, it is imperative to identify and disseminate the full range of evidence-based information of this form of technology. Such evidence will enable people to discuss the possible solutions offered by SAHRs,

in a more measured and informed way. This is particularly important in our days, since public attitudes towards robots may be also influenced by the media, often in negative ways. As an instance, while the use of robots will undeniably change the workforce, many people believe that these changes will only be negative. Example of catastrophic depictions of the use of AI in health and social care are that robots will take over human professionals' jobs, that robots will be dangerous, or that they are incapable of providing care that is culturally appropriate and compassionate.[16–19] In fact, the McKinsey Global Institute, along with a recent analysis led by PricewaterhouseCoopers, revealed that 'smart automation' that uses AI and robotics will be disruptive for many industries; yet some industries will be affected more than others. For example, in transportation and financial industries many low skills jobs that require repetitive tasks will be heavily affected. On the other hand, the healthcare sector will neither be affected in the same magnitude nor in a similar way.[20] Overall it is estimated that about 75 million to 375 million workers will have to change their occupation by 2030.[21] In the USA, employees in manufacturing, retail and accounting appeared more worried that AI would impact their jobs, whereas teachers, doctors and nurses were less so.[22] The Topol Review—that focuses on how the UK National Health Service (NHS) needs to prepare for the digital revolution—projects that, over the next 20 years, 90% of all NHS jobs will require the handling of data and the need for some digital skills.[23] The healthcare workforce will need to be educated in digital literacy according to their professional role, and new roles will be created as well. Similarly, in other industries it is projected that AI requires very specialised skills, and therefore the need for new technical jobs will increase in order to use robots in practice.[22 24] However, at least in the healthcare sector, nurses and other health professionals are seen working along with robots.[25] It is estimated that about 8%–16% of nursing time is consumed on a variety of non-nursing tasks that could be delegated.[26] Using robots for such tasks could free nurses' time to be spent in patient care.

### Objective

Our review aims to understand what the current enablers and barriers to the use and implementation of SAHRs are, and concentrates on articles that describe the actual use of SAHRs among older adults. The primary focus is on exploring and identifying the factors that might facilitate or hinder the implementation of SAHRs in health and social care for older adults.

### METHODS

#### Information sources and search strategies

The search strategy was developed for MEDLINE with appropriate modifications to match the terminology used in other databases. Databases were searched between 9 April 2018 and 8 June 2018. In view of the recent adoption of this form of technology, we limited the search date to the previous ten years. Subject headings and free text

## Box 1  Core set of search terms

- ► 'socially assistive robot*' OR 'socially assist*' OR 'social assist*' AND robot*
- ► AND 'social care' OR 'home care services' OR 'home care' OR 'care home*' OR 'nursing home*' OR 'residential facilit*' OR 'assisted living facilit*' OR 'group home*' OR 'home* for the aged' OR 'community health services' OR 'self-help devices' OR self* AND care* AND management AND help OR 'social support' OR 'interpersonal relations' OR 'nursing care' OR 'point of care' OR 'aged care' OR 'activities of daily living' OR care* OR healthcare OR social*
- ► NOT Animals NOT Infant OR Child* OR pediatr* OR paediatr*

terms were used according to the specific requirements of each database. Box 1 presents full search strategy with search terms across the following bibliographic electronic databases: the Cochrane Central Register of Controlled Trials; 2017 MEDLINE via OVID; Embase via OVID; Science Citation Index; Cumulative Index to Nursing and Allied Health Literature; Latin American and Caribbean Health Sciences Information database; IEEE Xplore digital library; PsycINFO; Google Scholar; European Commission and Eurobarometer. We also conducted the following additional searches: ACM Digital Library; Computer Source Lecture Notes in Computer Science; Science Direct. In addition to traditional searching, reverse citation screenings of the reference lists of relevant articles (ie, including the key terms such as SAHRs and home care) and forward citations (articles which have cited the identified papers) were conducted. The references of eligible reports and key review articles were examined for other potentially relevant studies.

All records were uploaded into Rayyan software, a systematic review software, similar to Covidence,[27] for managing citations for title and abstract screening and study selection.[28] The software was used for the process of

de-duplicating, and independently exploring, screening abstracts and full texts, excluding and including studies based on pre-specified criteria. Any disagreements regarding eligibility were discussed, and, if required, a third researcher was consulted, and consensus reached. Figure 1 summarises the selection of studies in accordance with PRISMA guidelines.[29]

### Selection criteria

Studies that considered the application of SAHRs only (ie, not animal-like robots) in health and social care were included. These were not restricted to experimental designs (table 1). In view of the likelihood of a paucity of potentially eligible studies relevant to this clinical topic, we also considered observational, cohort, case-control and qualitative studies. Editorials, conference abstracts and opinion pieces were excluded. Only adult and older adult care settings were included (eg, long term, rehabilitation, inpatient and outpatient hospital care, community and social care). The target population covered all stakeholders who were part of the process of implementation of SAHRs in health and social care in the broadest perspective (eg, users, staff, caregivers), and it was not limited to the aged population. Studies that included any type of direct exposure to SAHRs were selected.

### Data extraction and synthesis of the results

Study details and outcome data were collected independently by two researchers with a piloted data extraction form (see online supplementary file 1). The process was validated by assessing the data extraction form on a small number of studies (n=4) that two researchers assessed independently and compared. Type of study/design, date of publication, country and specific setting (ie, care facility), intervention (ie, type of SAHR), sample and characteristics of participants, and primary outcomes were identified (table 2). Primary outcomes entailed the identification of enablers and barriers to the implementation of SAHRs in health and social care. Barriers were defined as those impeding the implementation of SAHRs which may include factors, issues or themes at local, system or policy level. Enablers were defined as mechanisms and initiatives whereby patients, providers or policy makers contribute to facilitating the positive uptake and implementation of a SAHR.

The heterogeneity of the studies included in this review did not enable a standard quantitative synthesis (ie, meta-analysis) to be performed. Instead, a narrative synthesis of the results was conducted and presented in the form of a summary table (table 2) and figure (figure 2). All results were discussed and weighted by three researchers with the aim of identifying a frequency-based ranking of importance in relation to enablers, barriers and mixed results. Any uncertainties were resolved via a consensus-based decision. The protocol for this systematic review has been registered and published on PROSPERO.

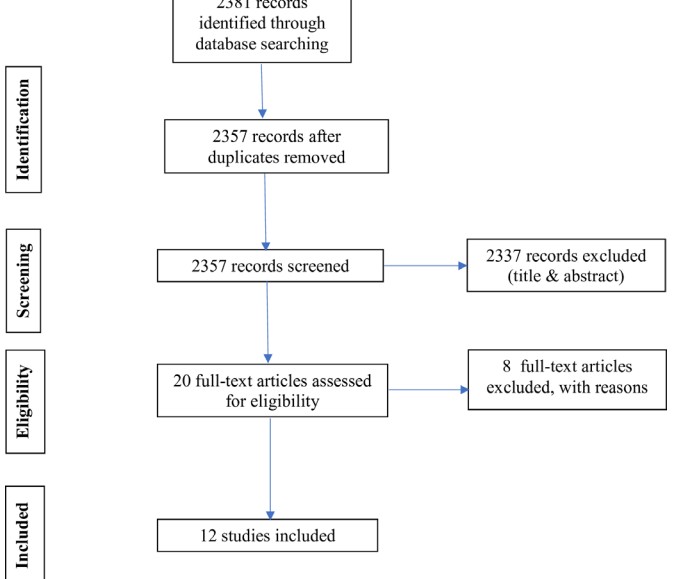

**Figure 1**  PRISMA flow chart.

**Table 1** Risk of bias and quality assessment of included studies

| | Authors year | Selection bias random sequence generation* | Allocation concealment | Performance bias blinding of participants and personnel | Detection bias blinding of outcome assessment | Attrition bias incomplete outcome data | Reporting bias selective reporting | Other bias and limitations† |
|---|---|---|---|---|---|---|---|---|
| 1 | Bedat et al[30] (2017) | − | − | − | − | − | − | Self-reported measures; no fidelity checks reported; small sample; gender imbalance; two-time interaction; not real home |
| 2 | Beuscher et al[36] (2017) | − | − | − | − | − | − | Self-reported measures; self-selected, very small, and WEIRD sample; no fidelity checks reported; SAHR small in size; one-time interaction |
| 3 | Caleb-Solly et al[31] (2018) | − | − | − | − | − | − | Not clear statistics; sampling unclear; small sample; no ethics reported; no fidelity checks reported |
| 4 | Hebesberger et al[32] (2017) | − | − | − | − | − | − | Sampling unclear; small sample; lack of validity of both qualitative and quantitative measures; low return on missing data |
| 5 | Khosla et al[38] (2017) | − | − | − | − | − | + | Not same cohort and not same RCF throughout the study; no fidelity checks reported; no ethics reported |
| 6 | Loi et al[37] (2017) | − | − | − | − | − | − | Self-reported measures; very small sample; low response rate; one-group design; large drop out at follow-up; exposure poorly measured; no fidelity checks reported |
| 7 | Louie et al[39] (2014) | − | − | − | − | − | + | Self-reported measures; sampling unclear; small, gender-imbalanced sample; low response rate; no ethics reported |
| 8 | Piezzo et al[40] (2017) | − | − | − | − | + | − | Sampling unclear; very small sample; no baseline data; no fidelity check; ethical approval not reported (only informed consent) |
| 9 | Sabelli et al[41] (2011) | − | − | − | − | − | − | Qualitative study; no comparator; no baseline; no confounders considered |
| 10 | Torta et al[34] (2014) | − | − | − | − | − | − | Self-reported measures; sampling unclear; small sample; low return on missing data; ethical approval not reported (only informed consent) |
| 11 | Werner et al[35] (2012) | − | − | − | − | − | − | Sampling unclear; small sample; no comparator; baseline data not reported; ethical approval not reported (only informed consent); no fidelity checks reported |
| 12 | Wu et al[33] (2014) | − | − | − | − | + | + | Partly self-selected sample; no fidelity checks reported |

*None of the studies was an RCT: therefore no randomisation was present. This also affects general quality of the studies, which overall were at high risk of all types of bias, with some exception in attrition and reporting bias only.

†Incorporation of evaluations conducted with critical appraisal for public health checklist (Heller et al 2007).

−, high risk of bias; +, low risk of bias; RCF, residential care facility; RCT, randomised controlled trial; SAHR, socially assistive humanoid robot; WEIRD, Western Educate Industrialised Rich Democratic.

**Table 2** Summary table of included studies

| | Authors year | Aim | Participants and sampling | Methodology and data collection | Intervention | Findings related to enablers (E) and barriers (B) |
|---|---|---|---|---|---|---|
| 1 | Bedaf et al[30] (2017) | Capture the experience of living with a robot at home | Aged 60+ (μ=78.9) participants living at home in the Netherlands, with no cognitive decline and receiving home care (n=10), informal caregivers (n=7) and professional caregivers (n=11). Non-probability convenience sampling | Mixed-method, no comparator, no baseline. Questionnaire and semi-structured interviews | Care-O-bot 3. Two-part-scenario, highly structured user test administered twice to each participant (preceded by a practise session). Duration of user test session: 1.5 hour | E: previous experience with technology; enjoyable experience; support to prolong independent living; tailored care; enhanced intelligence and social skills<br>B: technical problems; limited performance; lack of social interaction; arbitrary representations (eg, changing colour of robot torso) |
| 2 | Beuscher et al[36] (2017) | Determine impact of exposure to robots on perceptions and attitudes | Age 65+ (μ=81.9) participants with corrected vision and hearing that allowed them to engage in conversation with the SAHR, and physically able to participate in chair exercises (n=19). Non-probability convenience sampling | Pre-post intervention survey. No comparator. The 32-item acceptance scale which measured: performance expectancy, effort expectancy and attitudes. | NAO. Two sets of HRI experiments in an engineering lab (USA). The first set comprised of robot to one older adult, the second of robot to two older adults | E: familiarisation; higher education; enjoyable and engaging experience; easiness to understand SAHR; SAHR's pleasant appearance<br>B: life-like appearance (1/3 liked it); not feeling comfortable during HRI |
| 3 | Caleb-Solly et al[31] (2018) | Identify usability and user experience issues and how to overcome them | Aged 60+ (μ=79) group suffering from some ageing-related impairments but with stamina to participate in 2–3hour studies over a 5–6week period (n=11). Non-probability convenience sampling | Mixed-method, no comparator, no baseline. Questionnaire, structured interview, user experience analysis software, researchers' observations | Kompaï (Molly in the UK, Max and Charley in the Netherlands) physical robotic unit. Exposure: orientation workshops, individual trials in assisted living studio, residential care home user experience trial, home apartment user experience trial. | E: trust; familiarisation; cooperative interaction approach (co-learning self-training system); creative and engaging ways; use of Wizard; individualisation and contextualisation<br>B: technical problems |
| 4 | Hebesberger et al[32] (2017) | Investigate acceptance and experience of a long-term SAHR in a non-controlled, real-life setting | Staff members caring for older adults affected by dementia in care-hospital (Austria). | Mixed-method, no comparator, no baseline. Ten semi-structured interviews, live observations and n=70 online questionnaires | SCITOS robotic platform. 15-day trial following a 5-day pilot test | E: Mixed results on social acceptance; seen as a nice distraction, a novelty tool<br>B: technical problems; robot's lack of capabilities; negative views (robots vs humans); fear with new technology and with making mistakes |
| 5 | Khosla et al[38] (2017) | Study engagement and acceptability of SAHR among people with dementia | Aged 65–90 (μ=77.5) home care residents, with dementia and other conditions, living in residential care facilities in Australia (n=115). Total reactions coded and analysed: n=8304. Non-probability convenience sampling | Mixed-method, longitudinal experience trial. Video coding following engagement measures (emotional, visual, behavioural, verbal) during trial. Post-trial survey (acceptability based on TAM) | Matilda robotic unit. Designed activities in Matilda relevant to social context in RCFs in Australia. Repeated three-stage, 4–6hours long, field trials in four residential care facilities | E: services' personalisation (eg, songs and lyrics, integrated with human-like emotive expressions) accounting for users' disabilities; human-like features; personalisation underpinned in concept of personhood<br>B: technology barrier can be broken by accounting for the context of service, robot interface and users. |

Continued

**Table 2** Continued

| | Authors year | Aim | Participants and sampling | Methodology and data collection | Intervention | Findings related to enablers (E) and barriers (B) |
|---|---|---|---|---|---|---|
| 6 | Loi et al[37] (2017) | Investigate SAHR acceptance and utilisation | Staff in a residential care facility for younger adults in Australia. Pre-questionnaire (n=24) Post-questionnaire (n=8). Non-probability convenience sampling | Pre-post intervention survey. No comparator. TAM informed questionnaire with six statements pertaining to the staff themselves and 11 statements about the residents. Two post-questionnaire questions about the benefits and barriers | Betty. Two 1-hour long training/ introductory sessions over 2 weeks. In addition, Betty spent 12 weeks at the residential care facility | E: individualisation; music; the enjoyment interacting with the robot, being comfortable with the robot, perceived usefulness that the robot will improve their daily life and well-being; perceived beneficial for the patients B: excessive workload; negative assumptions on older residents' ICT skills; technical problems |
| 7 | Louie et al[39] (2014) | Explore acceptance and attitude towards human-like expressive SAHRs | n=54 older adults from a senior association (in Canada) (μ=76.5). Non-probability convenience sampling | Post-experimental survey. No comparator. TAM informed questionnaire of 18 items measuring seven constructs (n=46 completed the questionnaire, of which n=37 females) | Brian 2.1. One 1.5-hour-long live demonstration following person-centred behaviours guidelines | E: human-like communication is preferred over human-like appearance; gender (the male robot is more appreciated by female user) B: Feeling anxious with new technology *No relationship between previous experience and ease to use |
| 8 | Piezzo et al[40] (2017) | A feasibility study to assess the use of SAHR as a walking motivational partner | n=8 older adults with no cognitive problems (μ=82.5) visiting a facility that provides short-term care in Japan. Non-probability convenience sampling | Post-experiment motivation questionnaire (intrinsic motivation inventory) | Pepper used as a motivational walking partner. Older adults were asked to walk a short distance once on their own and once with the SAHR | E: personalised interaction; SAHR's encouragement and positive comments |
| 9 | Sabelli et al[41] (2011) | Unveil the experience of older adults and staff with a SAHR | n=55 cognitive healthy older adults (μ=83.9) visiting a elderly care centre in Japan either once or twice a week; n=8 female staff members. Non-probability convenience sampling | Qualitative study based on ethnography (semi-structured interviews, transcription of interaction, observations). Grounded theory used for data analysis. | Robovie2 placed for 3.5 months in an elderly day care centre. Robot teleoperated to engage in greetings and conversations | E: basic social interactions (ie, greetings and being called by name) sharing the routine; conversations about personal issues; SAHR's kindness and encouragement leading to positive emotions; attributed role as a child to the SAHR; staff positive attitudes and actions to favour HRI; Japanese culture B: SAHR's limited mobility and voice volume |
| 10 | Torta et al[34] (2014) | Investigate SAHR acceptance | n=16 older adults, cognitively healthy and able to perform physical exercises in a sitting position (μ=77), recruited from two senior citizen centres in Austria and Israel; ICT savvy older adults excluded. Non-probability convenience sampling | Repeated post-experimental survey. No comparator. Almere Model informed questionnaire. Repeated post-trial de-briefing interview for qualitative analysis | NAO as communication interface with KSERA smart home system. Short/long-term field trials involving five scenarios, totalling 22 trial iterations | E: small anthropomorphic shape (low anxiety); adaptability to user's needs/ personalisation; constant verbal communication; familiarisation (ease of use and sociability) B: small anthropomorphic shape (low social presence); technical malfunctions; familiarisation (as enjoyment linked to novelty effect) *No relationship between acceptance and user's cultural background |

Continued

 Papadopoulos I, et al. BMJ Open 2020;10:e033096. doi:10.1136/bmjopen-2019-033096

**Table 2** Continued

| | Authors year | Aim | Participants and sampling | Methodology and data collection | Intervention | Findings related to enablers (E) and barriers (B) |
|---|---|---|---|---|---|---|
| 11 | Werner et al[35] (2012) | Evaluate HRI and user experience | n=16 older adults, cognitively healthy and able to perform basic physical exercise (μ=77), recruited from two senior citizen centres in Austria and Israel. Non-probability convenience sampling | Post-test questionnaires containing KSERA HRI-Scale, the HRI Godspeed questionnaire, and questions regarding user acceptance. Notes on users' loud observations during test cases. Pre-test PANAS scales to evaluate participants' emotional state | NAO as communication interface with KSERA smart home system. Three test cases demonstrated twice to users | **E:** SAHR as motivator and helper; safety; sympathy, friendliness, intelligence **B:** technical problems (eg, navigation and speech recognition); limited performance negatively affecting human-likeness (eg, movement, navigation and conversational abilities) |
| 12 | Wu et al[33] (2014) | Investigate SAHR acceptance | Older adults (μ=79.3) with MCI and no impairment (n=11) in France. Non-probability convenience sampling | Mixed-method. Healthy group compared with group with MCI. No baseline, no treatment-as-usual comparator. Usability-performance measures, TAM informed acceptance questionnaire, semi-structured interview and focus group | Kompaï. Participants interacted with SAHR in the Living Lab once a week for 4 weeks. Duration of interaction: 1 hour | **E:** usability and amusement; encouragement to use from formal and informal carers professionals; sense of discovery and being up to date with technology; familiarisation **B:** uneasiness with technology; feeling of stigmatisation (ie, dependency and decline); ethical/societal issues associated with robot use (ie, fear of societal dehumanisation /changing human nature and what it means to care) *MCI may encounter more difficulties; CIH tended to show less positive attitude |

CIH, cognitively intact healthy; HRI, human–robot interaction; ICT, information and communications technology; MCI, mild cognitive impairment; RCF, residential care facility; SAHR, socially assistive humanoid robot; TAM, technology acceptance model.

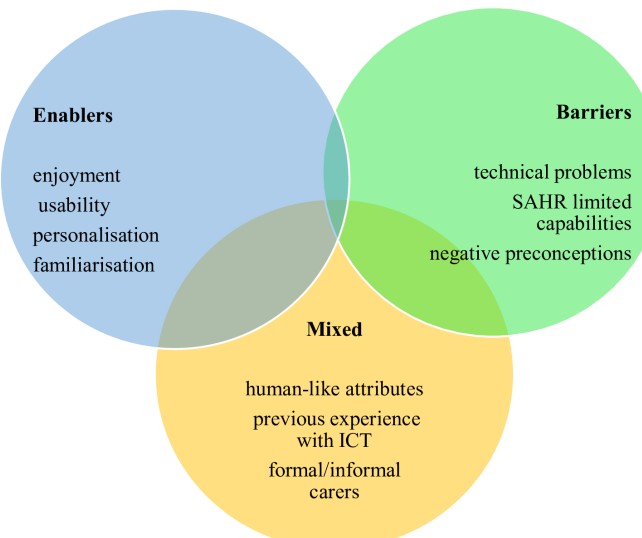

**Figure 2** Summary of results. ICT, information and communications technology; SAHR, socially assistive humanoid robots.

## RESULTS

### Search results and included studies

A total of 12 studies were included in our analyses: 6 mixed-method, non-randomised user experience trials[30–35]; 2 pre-post experimental surveys[36 37]; 1 mixed-method, longitudinal experience trial[38]; 2 post-experimental surveys[39 40]; and 1 ethnographic study[41] (see Figure 1).

### Assessment of risk of bias and quality of included studies

The quality of studies was assessed for all included studies with the following two assessments tools: the Cochrane Collaboration's tool for assessing risk of bias[42] and the critical appraisal for public health[43] (table 1). The research team decided that two researchers independently assessed four (ie, 1/3) of included studies and compared their results in order to ensure the validity and reliability of the process. Disagreements were resolved via the involvement of a third member of the research team and group discussions.

None of the selected studies had an experimental design; hence overall quality was low with high risks of biases (table 1). Most studies had no comparator and no baseline.[30–35 38 40 41] Additional methodological limitations affecting the non-randomised, quasi-experimental design of the studies were: very small samples' sizes, with only one study involving more than 100 participants[38]; and self-reported measures,[30 34–37 39 40] not always in combination with observation and/or data retrieved from the robot.[31–33 38] Seven studies[33–35 37 38 40 41] used validated instruments informed by existing theoretical models[44–47]; two studies reported the drop-out rate but did not mention the handling of missing data.[32 37] Three studies did not report any information on ethical approvals or consent received from the participants.[31 38 39] Protocols, trial pre-registration and fidelity checks were not found in any of the studies. Four studies reported no information about funding.[33 35 38 40]

### Characteristics of selected studies

Table 2 presents characteristics and outcomes of the 12 included studies.

### Population

Post-experimental data were collected and analysed for a total of 420 participants, including 73% of older adults (n=307), 2% of informal carers (ie, older adults' children, n=7) and 31% of formal caregivers and staff (n=106). The cohort of participants in two of the selected studies was the same[34 35]; however we resolved to count participants twice because aims, measures and results of the two studies were different. In 11 of the 12 selected studies, participants were older adults aged ≥60, with an overall mean age of 79.8 years. Among these 11 studies, 1 also included professional and informal caregivers,[30] and 2 considered residential care facility (RCF) staff.[32 41] One study only involved staff in a RCF for younger adults affected by neuropsychiatric conditions.[37] Three studies included older adults affected by dementia and other conditions of ageing-related, cognitive impairment.[31 32 38] One study compared older adults affected by mild cognitive impairment with a cognitively intact healthy (CIH) group,[33] whereas another one did not compare the two groups.[36] Five studies selected CIH older adults,[30 34 35 40 41] whereas in another one participants' condition was not reported.[39] Since one study did not report the gender of the 55 older adults taking part in the study,[41] out of 365 participants, 69% were women. Participants' level of education was only considered in three studies where over 80% of participants had at least a bachelor's degree.[33 36 39]

Similarly, in the four studies where data were collected on general information and communications technology (ICT) skills, 76% (n=66) of 87 participants reported regular computer use.[31 33 36 39] In other two studies,[34 35] highly experienced technology users were excluded, following assessment. In these two studies, information around previous contact with a SAHR on behalf of research participants is not explicit. However, if we assume that high ICT experience implies previous contact with a SAHR, none of the participants across all the studies had had any hands-on experience with SAHRs before taking part in the studies.

The largest post-experimental group consisted in Australian participants (n=123). All the earlier figures includes neither data of subjects who dropped out in pre-post studies[37 39] nor all data collected via observation of HRIs or interviews, as sometimes this information was irretrievable or not reported.[32]

### Settings and interventions

Four trials were carried out in RCFs,[32 37 38 41] six in smart environments or university laboratories,[30 33–36 39] and two in a combination of private apartment, RCF and laboratory.[31 40] None of the studies was conducted in an acute healthcare setting. Studies were conducted in the following countries: six in a European context (Austria, UK, Netherlands, France);[30–35] and two of these six in

Israel as well;[34 35] two in Australia;[37 38] two in Japan;[40 41] one in Canada;[39] and one in the USA.[36]

All studies included interventions where participants had their first hands-on experience interacting with a SAHR. Eight different types of SAHRs were used which had different appearances, bodily movements' abilities, often an additional mode of interaction beyond voice-based (ie, built-in touch screen, touch sensors, tablet remote control). All were customised with software packages providing a range of specific services.

In most studies, a pilot field test was conducted to establish familiarisation. Pilot testing was deemed necessary particularly in those experiments where participants had to interact with the SAHR in highly structured scenarios performing specific tasks, sometimes following instructions.[30 31 33–36 40] This type of HRI lasted between 45 min[30] and up to 6 hours.[38] Three studies adopted a design whereby HRI was not structured, and RCFs residents and members of staff freely chose to interact with the SAHR.[32 37 41]

The HRIs in the 12 exposures involved the following services and activities: playing cognitive games such as Bingo, Hoy and general knowledge games, including an orientation game with the support of pictures, '21 questions' and 'Simon says' game;[31 33 36–39] listening to music, singing, storytelling, relaxation, dancing (including joint chair exercise) and physical training (including walking);[34–38 40 48] carry and delivery tasks;[30 31] call to a friend, calendar and reminders such as to drink water, to do exercise, to take medication;[30 31 33 34] weather information;[34 37] restaurant finding;[39] and reception, greetings and interactions;[32 41] medical measurement.[35]

### Narrative synthesis

Findings in terms of enablers and barriers are presented below and summarised in figure 2.

### Enablers
#### *Enjoyment*

An enjoyable experience was found to be a crucial factor conducive to SAHR's use and implementation. In ten trials (83%) participants highly valued enjoyment and engagement when interacting with the SAHR, both in terms of general positive HRI experience (eg, SAHR's kindness, friendliness, provision of comfort and motivation) and in relation to specific activities (eg, listening to music and playing games). In one study only,[34] it is reported that participants to the long-term trials of the intervention commented negatively with respect to their enjoyment in interacting with the robot, and furthermore that this would decrease over time.

#### *Usability*

Intuitiveness and easiness of use proved to be essential enablers towards the implementation of SAHRs in six studies (50%).[30 31 33 34 36 38] Usability is to be broadly intended in terms of lack of technical issues, intuitive interface and design factoring participants' disabilities.

#### *Personalisation*

Engagement and enjoyment were found to be interlinked with the personalisation of services, hence ultimately with overall use and implementation. Personalisation should account for: adaptation to users' taste and preferences;[38 40] user's care needs,[30] context and routine;[31 41] and users' impairments.[33 38 41]

#### *Familiarisation*

Inasmuch as the robot should offer individualised services, users also should learn about and adapt to the robot's status and intentions.[31] While the model of human–robot co-dependent relationship is prominent in one study only,[31] other studies found familiarisation to be an important factor positively affecting implementation.[33 34 36 38] Interestingly, in one of these studies participants felt that not only over time ease of use would improve, but also that the relationship with the SAHR may turn into a friendship.[34]

### Barriers
#### *Technical problems*

Over half of the studies[30–32 34 35 37 41] explicitly stated that technical issues with the robot itself constituted a barrier to SAHR's implementation in health and social care.

#### *SAHR's limited capabilities*

The limited performance (ie, mobility, robots' voice, lack of interactive element) of the robot was found as a crucial barrier to use. This impediment was explicitly reported in four studies,[30 32 35 41] while more implicitly in other three, where the robot's restricted skills were described in terms of limited personalisation of services,[38] adaptability[34] and co-learning/self-training abilities.[31]

#### *Negative preconceptions*

In a study, health professionals' assumptions on older adults' capacity to interact with SAHR were included among the barriers to implementation.[37] Two other studies elaborated on the negative views towards robots in terms of dehumanisation of care and society,[32 33] and of stigmatising effects associated to being a dependent individual in decline.[33] In three studies, negative preconceptions came from formal and informal carers rather than from older adults themselves.[30 32 37]

### Mixed views
#### *Human-like attributes*

One study showed that human-like appearance was appreciated by one-third of the participants.[36] Another study reported that human-like communication was preferred over human-like appearance.[39] In the same study, 80% of the subjects completing the trial were older women who declared to prefer a male looking SAHR with male voice.[39] A third study concluded that SAHRs based on human-centred system with human-like characteristic are likely to enable acceptance and use.[38] However, in the same study, it was also reported the fact that the SAHR was not judgemental facilitated interaction.[38] The ambivalence

of having a non-judgmental conversational partner (ie, non-human) who was also given the overt social role of a human child was found beneficial to implementation in a fourth study.[41] SAHR's child-likeness was also found positive in a fifth study, and SAHR's small size was appreciated, although contributing to reduced acceptance with low scores in attributed animacy and naturalness.[35] Similar ambivalent results are found in a sixth study where again SAHR's small anthropomorphic shape was at the same time responsible for low levels anxiety, but also for low scores in perceived social presence.[34] In relation to social presence participants had contrasting views (ie, SAHR seen as pet or a conversational partner). Differently from these last studies, in a seventh one participant did not choose to walk side-by-side with the SAHR, as it would be natural with a human partner, but chose to follow the SAHR, giving it the role of a guide.[40] Finally, in an eighth study, the lack of more complex social interaction was identified as a barrier to implementation.[30] None of the other studies provided any indication regarding the cultural attributes of the SAHR. In one study only, it is reported that the fact that the SAHR was speaking the same language of the users was responsible of higher perceived ease of use compared with the cohort where the SAHR was not using the users' native language.[34] In another study, it was argued that the positive reception of the robot may be also attributed to the nature of the local culture (ie, Japanese) towards robots.[41]

### Previous experience with ICT

While one study found that previous experience with technology positively correlated with use,[30] another trial found that there was no relationship between previous experience and ease to use.[39] In other two studies, highly experienced ICT users were excluded from participating in light of the argument that acceptance is positively influenced by ICT experience.[34 35]

### The role of formal and informal caregivers

As mentioned earlier, the negative attitudes of formal and informal carers have been shown to constitute an impediment to SAHR's implementation.[30 32 37] Conversely, two studies highlighted the enabling effect of the encouragement for SAHR's use on behalf of relatives and professionals.[33 41]

## DISCUSSION
### Summary of evidence

Our review focused on the identification of factors that could facilitate or hinder the implementation of SAHRs in health and social care. We focused on actual interactions of older adults with social humanoid robots in different settings, in order to better understand what the current issues are in regard to implementation. Enablers, such as enjoyment and personalisation, were mainly related to the use of robots at an individual level. The element of enjoyment in the HRI was also elsewhere

found to be crucial among hospital patients,[49] opening the doors for considering social humanoid robots as an intervention to combat social isolation in hospital settings.

Barriers were related to technical problems and to current limited capabilities of the robots. Technology malfunction and/or technology limitations were reported as areas of concern, similar to the results of a recent survey of Korean nurses.[50] Surprisingly for the heavily regulated field of healthcare, the issues of safety, ethics and safeguarding were not identified in this review as significant implementation-related factors, even though nurses and healthcare workers have been raising these issues. Safety and ethical issues were reported as major concerns in previous systematic reviews, and it is imperative that future research investigates these issues and understands their implications. The field of social humanoid robots poses many ethical challenges especially because robots could be designed to assume different roles and for different purposes: from service robots assisting in concierge types jobs to companion robots. In agreement with Vandemeulebroucke et al,[51] we believe that an ethical approach demands that all stakeholders should have a voice in the current debate, but also in the design of future technologies, their application and implementation. We also agree with Chou et al[52] that future planning should view all these factors under a broader policy framework, and policy makers should work collaborative to ensure the ethical and safe implementation of robots. The European Commission advocates for the use of a new framework to address the ethical issues in healthcare robotics called 'Responsible Research and Innovation'.[53] Under this framework, society, users and innovators are mutually responsive and engage in an interactive and transparent process in order for acceptable, sustainable and desirable products to be developed and embedded in our society. Similarly, the Alan Turing Institute calls for the use of a framework of ethical values that need to guide every AI project, and they introduce the use of four actionable principles: (i) fairness, (ii) accountability, (iii) sustainability and (iv) transparency.[54] These principles are reflected onto the current UK code of conduct for data-driven health and care technology,[55] and onto the current policy paper for the safe and ethical introduction of AI in the NHS.[56] Fairness refers to the avoidance of bias and discrimination, for example, and according to it, the AI system should use only fair and equitable data. Accountability refers instead to the auditability of the system, ensuring that responsibility of all actions is established throughout the AI system, from the design to the final implementation. Sustainability of the system refers to the safety, reliability, accuracy and robustness of the system. Finally, transparency covers the ability of the designers to always explain how the system is working and how it will affect its users. Ensuring the use of ethical guidelines in the design of AI and robotics interventions is critical since many interventions are still designed without the consideration of ethics.[57]

Robot's appearance[30 36 38 39] and views of carers and relatives provided mixed results.[30 32 33 37] In regard to the appearance, Mori's theory of the 'uncanny valley' is illuminating.[58] Between the animated and the perfectly realistic, human-like appearance of robots, there is an area where depictions can create uncomfortable feelings in humans. Therefore, life-like attributes of the robots, such as voice, facial expressions, gestures, bodily appearance, cultural attributes and gender, have an impact on how the user experiences the robot, and on the HRI. The indeterminacy of robots' appearance is reflected onto the dramatic variations of SAHRs found in the literature. We also know that one's cultural background influences views and perceptions of the robot's aesthetics,[11] but none of the studies provided any indication regarding the cultural attributes of the SAHR. Culturally specific research on the relationship between appearance, acceptance and implementation is therefore promising in HRI studies.

According to our protocol, we searched for factors affecting the implementation of SAHRs by key stakeholders, such as health professionals. The role of formal and informal caregivers has been found as crucial.[59] However, the information we could yield was limited and mixed, and this is an area that urgently requires further research, involving longitudinal studies and larger samples. Longitudinal studies can provide the opportunity to investigate whether fear of using a new and unfamiliar technology, or losing interest in a new technology (diminishing novelty effect), are related to negative attitudes. Abbott *et al*[8] in their review of the use of social robotic pets (animal-like social robots) found similar mixed feelings from the different stakeholders. The fact that people have very strong feelings on the opposite sides of the spectrum, either very positive or very negative, is significant to implementation and requires a careful investigation. The current Topol Review[23] addresses the changes and accompanied needs of the healthcare workforce that will be imposed by the digital revolution. It calls for an urgent need to educate and prepare the healthcare workforce for the imminent digital changes and for an organisational cultural change. However, it is hard to think how these transformations will happen when the current evidence reveals the existence of mixed opinions and negative attitudes at least towards the use of socially assistive robotic technologies.

The completeness and overall applicability of the evidence are limited, mostly because it provides only insights into individual-level factors related to the acceptance of technology. This can be partly attributed to the main theoretical framework used in the studies. The technology acceptance model (TAM) proposes in fact an explanation for a person's actual and intentional use of a technology, through an exploration of their attitudes towards it.[44] The lack of evidence related to other main key stakeholders, such as formal and informal carers, along with factors related to the environment, policy, society and organisation is a major limitation. Exploring attitudes of other populations, such as formal caregivers,

as well as the use of other theoretical models, is considered critical. The field would benefit, for example, from the use of the diffusion of innovations theory (DIT),[60] when considering research questions related to the use of SAHRs in healthcare; but also from theories that explore the co-existence of technology and caring, such as the theory of technological competency as caring in nursing.[61] King and Barry[62] recently introduced a theoretical model that highlights caring theories when considering the design of healthcare robots. Understanding how nursing care will change, or what will be the best interface of nurses with SAHRs is critical. In addition, how compassionate care will be understood, expressed and studied is also essential. The Papdopoulos model that integrates compassion into culturally competent care would be useful in exploring the interrelations between service users, nurses, health professionals, family members and SAHRs.[63] Furthermore, researchers working in the area of HRI among older adults are calling for new ways to conceptualise ageing and consequently robotic technologies. In particular, they advocate that the use of socially assistive robots should be studied under a model that focuses on 'successful aging' rather than a 'deficit model of aging'. They argue that the latter model—viewing ageing a process of continued losses and older adults needing assistance—restricts the design of new technologies. A successful model of ageing that focuses on the preservation of the user's autonomy can instead provide new ways of using, designing and implementing socially assistive robots.[64]

### Limitations

As per protocol, our intention was to explore enablers and barriers to the implementation of SAHRs in both health and social care but, in fact, most of the activities assessed were more relevant to social care. Even medication reminders, which are obviously health-related, form an important part of social care. There is therefore little to inform health practitioners as to the possible application of SAHRs in health settings. Furthermore, very few studies have deployed and implemented SAHRs in health and social care settings; hence the available information is scant. In addition, quality of the studies is problematic (table 1).

The heterogeneity of study designs led to the identification of factors in single studies. For example, only one study reported on the level of education as enabling factor of SAHR's acceptance.[36] Another study found that fear of making mistakes with technology was a barrier to implementation.[32] However, in another study, uneasiness with technology seemed to be counterbalanced by a sense of discovery and being up-to-date with ICT.[33] The evidence is too scant to generalise these initial findings, and further research is needed to assess the impact of these, and other factors, onto SAHR's acceptance and implementation in health and social care.

## CONCLUSION AND PERSPECTIVES

The use of SAHRs is promising in responding to some of the care challenges of an ageing population. This systematic review summarised the enablers and barriers to the implementation of SAHRs in health and social care. Evidence suggests that enjoyment and personalisation are the chief enablers to the implementation of robots, while the two most important barriers had to do with technical problems and the limited capabilities of the robots. However, there are limitations to the evidence, as most studies were at high risk of bias involving very small samples. Gaps in the evidence include factors related to environment, organisation, socio-cultural milieu, policy and legal framework. Furthermore, the research focus has currently been placed on understanding the acceptance of robots by adult users, but there is no discussion of the needs of the healthcare workforce on a professional level, and how these needs are being met by educational institutions, professional organisations, and employers.

**Acknowledgements** We would like to thank Dr Zbys Fedorowicz for his contributions and helpful advice.

**Contributors** IP and CK conceived and designed the study. CK and SA acquired, screened and conducted the initial data extraction. IP acted as a referee in the screening, analysis and interpretation of the data, which was conducted by CK and RL, who both drafted the manuscript, and IP provided the critical review of it. All authors approved the version to be published and agreed to be accountable for all the aspects of this work.

**Funding** This work was supported by CARESSES project (Horizon 2020, Grant Agreement ID: 737858).

**Competing interests** None declared.

**Patient consent for publication** Not required.

**Ethics approval** Ethics approval was not required for this research.

**Provenance and peer review** Not commissioned; externally peer reviewed.

**Data availability statement** Data are available upon reasonable request.

**ORCID iD**
Runa Lazzarino http://orcid.org/0000-0002-4206-4913

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
