## [Reviewer comments · BMJ Open]

ARTICLE DETAILS

TITLE (PROVISIONAL)	A systematic review of enablers and barriers to the implementation of Socially Assistive Humanoid Robots in health and social care
AUTHORS	Papadopoulos, Rena; Koulouglioti, Christina; Iazzarino, Runa; Ali, Sheila

VERSION 1 – REVIEW

REVIEWER	Fabrice Jotterand Medical College of Wisconsin, USA
REVIEW RETURNED	05-Aug-2019

GENERAL COMMENTS	This is an interesting and important review of the literature pertaining to the implementation of SAHRs in health and social care. The results demonstrate that more work on the implications of the use of SAHRs is needed for a responsible use of these technologies and harvest their benefits in the health care context. That said, I would encourage the authors to consider the following critical points: Specific comments: p. 5 line 17: "robots that were considered nearer to a person's culture..." - this statement is ambiguous and should be more fully developed. Nearer in what sense? p. 6 lines 14-23: the media are not the only source of influence. In addition, a more nuanced statement should be crafted concerning jobs - the reality is that AI will change the workforce. There is ample evidence that the authors should consider. p. 8 line 52: why 40%? p. 13 line 46: the issue of bias should be more carefully explained. What are the criteria defining bias in the context of these studies? Say more. p. 14 line 28: informal caregivers - who are these individuals? p. 15 I am confused as to whether participants had previous hands-on experience with SAHRs. Line 5 seems to indicate no but line 36 indicates the opposite. The authors might refer to different studies...please clarify.
---

	p. 22 lines 23ff: any indication about the "ethnic attributes" of the robot? Sensitive topic but I wonder whether such dimension was captured? line 47: "negative attitudes of formal and informal carers" - any suggestions how to address this issue? While it is not the purpose of this review it would be interesting to hear from the authors whether they have any concrete proposals. This could be developed in the Discussion section. p. 23 I understand the nature of reviews as this one and the purpose of the Discussion section. However, considering the nature of the issues discussed, this section could be expanded, especially since the authors mention "the need to ensure the ethical and safe implementation of robots". The discussion should be the basis for a follow-up article to address these implications. In other terms, the Discussion section is somewhat weak in its critical and conceptual dimensions.
--	---

REVIEWER	Hee Rin Lee Michigan State University
REVIEW RETURNED	23-Aug-2019

GENERAL COMMENTS	This manuscript provides a systematic review of Socially Assistive Robot (SAR) studies. The authors collected data from ten databases including MEDLINE, IEEE Xplore, and Google Scholar. The manuscript identifies existing factors that act to facilitate or hinder the adoption of SAR in health and social care. As a result, the authors suggest new SAR research directions. The main contribution of this manuscript is that it presents and analyzes existing SAR literature from the perspective of healthcare professionals. As the authors discuss, SAR research needs more input from healthcare researchers. For example, SAR research will be enriched by integrating theories from nursing and by investigating the relationships between SAR and the healthcare workforce on a professional level. However, this paper has a major methodological issue concerning the collection of existing SAR literature. Unlike other fields, the field of computer science publishes its research largely at conferences. These conference publications are thoroughly peer-reviewed and contain 10 page-long texts within a two-column and single-spaced format. These publications are equivalent to journal articles in this field. SAR emerged within the field of Human-Robot Interaction (HRI) and has been studied for more than a decade [1, 2, 3]. It is not true that there are only eight studies investigating SAR and the eight studies cited in the current manuscript do not adequately represent SAR research. I would strongly suggest that the authors include conference papers, which probably make up a larger proportion of the 2222 articles removed from the authors' initial data. In particular, Human-Robot Interaction (http://humanrobotinteraction.org/2020/) is a flagship conference in the HRI community and the authors should include the associated publications in their analysis, such as [4, 5, 6, 7]. Overall, this article is a well-written manuscript. However, it has a significant methodological issue that needs to be addressed to be suitable for publication.
--

	References [1] Tapus, A., Maja, M., & Scassellatti, B. (2007). The grand challenges in socially assistive robotics. [2] Feil-Seifer, D., & Mataric, M. J. (2005, June). Defining socially assistive robotics. In 9th International Conference on Rehabilitation Robotics, 2005. ICORR 2005. (pp. 465-468). IEEE. [3] Forlizzi, J., DiSalvo, C., & Gemperle, F. (2004). Assistive robotics and an ecology of elders living independently in their homes. Human-Computer Interaction, 19(1), 25-59. [4] Beer, J. M., Smarr, C. A., Chen, T. L., Prakash, A., Mitzner, T. L., Kemp, C. C., & Rogers, W. A. (2012, March). The domesticated robot: design guidelines for assisting older adults to age in place. In Proceedings of the seventh annual ACM/IEEE international conference on Human-Robot Interaction (pp. 335-342). ACM. [5] Chang, W. L., & Sabanovic, S. (2015, March). Interaction expands function: Social shaping of the therapeutic robot PARO in a nursing home. In 2015 10th ACM/IEEE International Conference on Human-Robot Interaction (HRI) (pp. 343-350). IEEE. [6] Sabelli, A. M., Kanda, T., & Hagita, N. (2011, March). A conversational robot in an elderly care center: an ethnographic study. In 2011 6th ACM/IEEE International Conference on Human-Robot Interaction (HRI) (pp. 37-44). IEEE. [7] Lee, H. R., & Riek, L. D. (2018). Reframing assistive robots to promote successful aging. ACM Transactions on Human-Robot Interaction (THRI), 7(1), 11.
--	--

REVIEWER	Zeraati H Tehran University of Medical Sciences, Iran
REVIEW RETURNED	28-Aug-2019
GENERAL COMMENTS	This MS is not a systematic review. It is only a literature review article, without using systematic review methodology.

VERSION 1 – AUTHOR RESPONSE

REVIEWER 1	
p. 5 line 17: "robots that were considered nearer to a person's culture..." - this statement is ambiguous and should be more fully developed. Nearer in what sense?	New sentence added to remove the ambiguity. Please, see marked copy, p. 5
p. 6 lines 14-23: the media are not the only source of influence. In addition, a more nuanced statement should be crafted concerning jobs - the reality is that AI will change the workforce. There is ample evidence that the authors should consider.	Change and additions made. Please, see marked copy, p.6. However, the aim of our statement is to highlight that often 'as an instance' the media which give voice to e.g. some politicians, policy makers, and representatives of institutions in social and health care, emphasise the storyline that 'robots will take over healthcare professionals jobs'. Ample evidence is considered with 5 pieces of literature quoted here.
p. 8 line 52: why 40%?	Please, see marked copy, p. 9-10. The research team decided that double assessment of studies was good standard measure to ensure validity and reliability of the process of quality

	assessment. With the new studies included, 4 studies have been analysed and assessed by two team members. This means 1/3 of the evidence selected was doubly assessed.
p. 13 line 46: the issue of bias should be more carefully explained. What are the criteria defining bias in the context of these studies? Say more.	Some changes made. Please, see marked copy, pp.10-13. Explanation and summary of factors which explain which the biases are in these studies is provided in the text. Thorough information are to be found in both table 2 and 3. In particular, criteria of bias for all studies appear as headings on table 2 (pp.10-12), e.g., Selection Bias – Random Sequence Generation. In terms of the definition of high and low risk of bias according to the Cochrane Collaboration's tool for assessing risk of bias used, the assessment was conducted based on the authors' judgment.
p. 14 line 28: informal caregivers - who are these individuals?	Info added, please see marked copy, p. 14. However, please note that the expression 'informal caregivers' is widely used and accepted, and it refers to the patient's family members, friends and volunteers, who are normally not paid to provide care.
p. 15 I am confused as to whether participants had previous hands-on experience with SAHRs. Line 5 seems to indicate no but line 36 indicates the opposite. The authors might refer to different studies...please clarify.	Clarification added, please see marked copy, pp. 20. Participants had no hands-on experience prior to the study, during which they had their first experience interacting with a humanoid robot.
p. 22 lines 23ff: any indication about the "ethnic attributes" of the robot? Sensitive topic but I wonder whether such dimension was captured?	Additions made on pp. 24 and in the discussion on p.26, please see marked copy. We normally however refer to culture and not ethnicity in this article. Thus on the pages indicated references to cultural background and attributes can be found.
line 47: " negative attitudes of formal and informal carers " - any suggestions how to address this issue? While it is not the purpose of this review it would be interesting to hear from the authors whether they have any concrete proposals. This could be developed in the Discussion section.	Additions were made in the discussion, please see marked copy on p. 26-27.
p. 23 I understand the nature of reviews as this one and the purpose of the Discussion section. However, considering the nature of the issues discussed, this section could be expanded, especially since the authors mention "the need to ensure the ethical and safe implementation of robots ". The discussion should be the basis for a follow-up article to address these implications. In other terms, the Discussion section is somewhat weak in its critical and conceptual dimensions.	Additions were made in the discussion, please see marked copy on p. 25 and 26.
REVIEWER 2	
However, this paper has a major methodological issue concerning the collection of existing SAR literature. Unlike other fields, the field of computer science publishes its research largely	Thank you very much for this comment. Following your suggestion, we further searched the HRI journal, we reviewed the articles you recommended, we searched again the ACM

at conferences. These conference publications are thoroughly peer-reviewed and contain 10 page-long texts within a two-column and single-spaced format. These publications are equivalent to journal articles in this field. SAR emerged within the field of Human-Robot Interaction (HRI) and has been studied for more than a decade [1, 2, 3]. It is not true that there are only eight studies investigating SAR and the eight studies cited in the current manuscript do not adequately represent SAR research. I would strongly suggest that the authors include conference papers, which probably make up a larger proportion of the 2222 articles removed from the authors' initial data. In particular, Human-Robot Interaction (http://humanrobotinteraction.org/2020/) is a flagship conference in the HRI community and the authors should include the associated publications in their analysis, such as [4, 5, 6, 7]. Overall, this article is a well-written manuscript. However, it has a significant methodological issue that needs to be addressed to be suitable for publication.	digital library and google scholar focusing only on articles describing the use of a socially assistive humanoid robot (SAHRs) – as these differ from socially assistive robots (SARs) during the decade 2008-2018. The retrieved articles were screened following our inclusion and exclusion criteria. This process resulted in 12 full-text articles that two of the authors read independently. Eight articles were excluded with reasons (e.g. studies published after June 8th 2018 or not providing any information on implementation) resulting in the inclusion of 4 new articles in our final review. The PRISMA figure has been revised accordingly. Specific answers to the recommended references:  1. Included in the introduction 2. Included in the introduction 3. Excluded (not in time-limits of this review) 4. Excluded (older adults watching a video not hands on interaction) 5. Excluded (related to an animal-like robot) 6. Included in the final selection of articles for review 7. Included in the discussion
REVIEWER 3	
This MS is not a systematic review. It is only a literature review article, without using systematic review methodology.	Prospero, the main registration institute for systematic reviews, accepted the protocol of this study as a systematic review and registered it on its database. Please, see also the Joanna Briggs definition of systematic review: "Systematic reviews aim to provide a comprehensive, unbiased synthesis of many relevant studies in a single document using rigorous and transparent methods. A systematic review aims to synthesize and summarize existing knowledge. It attempts to uncover "all" of the evidence relevant to a question. [...] Chalmers and Altman (1995) suggested that the term 'meta-analysis' be restricted to the process of statistical synthesis, that is meta-analysis may or may not be part of a systematic review. [...] There is general acceptance of the following steps being required in a systematic review of any evidence type. These include the following:  Formulating a review question Defining inclusion and exclusion criteria Locating studies through searching Selecting studies for inclusion Assessing the quality of studies Extracting data

	Analyzing and synthesizing the relevant studies Presenting and interpreting the results, potentially including a process to establish certainty in the body of evidence (through systems such as GRADE) An essential step in the early development of a systematic review is the development of a review protocol. A protocol pre-defines the objectives and methods of the systematic review which allows transparency of the process which in turns allows the reader to see how the findings and recommendations were arrived at. It must be done prior to conducting the systematic review as it is important in restricting the presence of reporting bias. The protocol is a completely separate document to the systematic review report.” (emphasis added)
--	---

VERSION 2 – REVIEW

REVIEWER	Fabrice Jotterand Medical College of Wisconsin - USA
REVIEW RETURNED	09-Oct-2019

GENERAL COMMENTS	This version of the paper has been greatly improved with regard to the methodology and the discussion/conclusion sections. However, I still think that there are few areas where the manuscript could be improved. In particular, 1) p. 6 issue related to the workforce. Providing clear data would help. In the US and Europe that there is evidence that overall the workforce will be impacted negatively. In addition, it is not clear that jobs requiring "very specialized" will offset the negative impact of the use of AI. 2) The discussion section RE ethical implications could still benefit from a deeper dive. The reference to the European commission document on Responsible Research and Innovation is a good start but elaborating a bit more on concepts such as transparency, responsiveness etc. would benefit the overall analysis,
---

VERSION 2 – AUTHOR RESPONSE

REVIEWER 1	
1) p. 6 issue related to the workforce. Providing clear data would help. In the US and Europe that there is evidence that overall the workforce will be impacted negatively. In addition, it is not clear that jobs requiring "very specialized" will offset the negative impact of the use of AI.	Thank you for the comment. An additional paragraph was added providing information related to estimated workforce changes.
2) The discussion section RE ethical implications could still benefit from a deeper dive. The reference to the European commission document on Responsible Research and	Thank you for the comment. An additional paragraph was added in this section as well, referencing all the latest policy papers and describing the main ethical guidelines for the AI

Innovation is a good start but elaborating a bit more on concepts such as transparency, responsiveness etc. would benefit the overall analysis,	implementation.
---	-----------------